



# A systematic assessment of water vapor products in the Arctic: from instantaneous measurements to monthly means

Susanne Crewell[1], Kerstin Ebell[1], Patrick Konjari[1], Mario Mech[1], Tatiana Nomokonova[1], Ana Radovan[1], David Strack[1], Arantxa M. Triana-Gómez[2], Stefan Noël[2], Raul Scarlat[2], Gunnar Spreen[2], Marion Maturilli[3], Annette Rinke[3], Irina Gorodetskaya[4], Carolina Viceto[4], Thomas August[5], and Marc Schröder[6]

[1]Institute for Geophysics and Meteorology, University of Cologne, Cologne, Germany
[2]Institute of Environmental Physics, University of Bremen, Germany
[3]Alfred Wegener Institute, Helmholtz Centre for Polar and Marine Research, Potsdam, Germany
[4]Centre for Environmental and Marine Sciences, University of Aveiro, Portugal
[5]Eumetsat, Darmstadt, Germany
[6]Deutscher Wetterdienst, Offenbach, Germany

**Correspondence:** Susanne Crewell (susanne.crewell@uni-koeln.de)

**Abstract.** Water vapor is an important component in the water and energy cycle of the Arctic. Especially in the light of Arctic amplification, changes of water vapor are of high interest but are difficult to observe due to the data sparsity of the region. The ACLOUD/PASCAL campaign performed in May/June 2017 in the Arctic North Atlantic sector offers the opportunity to investigate the quality of various satellite and reanalysis products. Compared to reference measurements at R/V *Polarstern*

frozen into the ice (around 82° N, 10° E) and at Ny-Ålesund, the Integrated Water Vapor (IWV) from IASI shows the best performance among all satellite products. Using all radiosonde stations within the region indicates some differences that might relate to different radiosonde types used. Though the region is well sampled by polar orbiting satellites daily means can deviate by up to 50 % due to strong spatio-temporal IWV variability associated with atmospheric river events. For monthly mean values, this weather induced variability cancels out but systematic differences dominate which particularly appear over

different surface types, e.g. ocean, sea ice. In the data sparse central Arctic above 84° N, strong differences of 30 % in IWV monthly means between satellite products occur in the month of June which likely results from the difficulties to consider the complex and changing surface characteristics of the melting ice within the retrieval algorithms. There is hope that the detailed surface characterization performed as part of the recently finished Multidisciplinary drifting Observatory for the Study of Arctic Climate (MOSAiC) will foster the improvement of future retrieval algorithms.

## 1 Introduction

Water vapor plays an important role in the hydrological cycle of the Arctic as a source for cloud and fog formation and by its effects on the energy budget via condensation/evaporation and radiative transfer (Vihma et al., 2016). For the Arctic





climate, water vapor is of particular interest as it could contribute to Arctic amplification by enhancing downwelling longwave
radiation (Ghatak and Miller, 2013) and providing the moisture source for precipitation. Investigating relative contributions of
the different feedback processes is still under debate and focus of current research (e.g., Wendisch et al., 2017; Serreze and
Barry, 2011).

Despite its importance, the Arctic suffers from a lack of reliable water vapor measurements due to the limited amount of
surface stations. Thus, studies aiming at the detection of changes in the water vapor distribution mainly make use of reanalyses
and radiosondes (e.g., Serreze et al., 2012; Dufour et al., 2016; Rinke et al., 2019). The latter are limited to land areas and
concentrated in northern America and Europe. Furthermore, measurements at low temperature are especially challenging and
the correction of sensor time lag and radiation dry bias are best addressed by using Global Climate Observing System (GCOS)
Reference Upper-Air Network (GRUAN) processing (Dirksen et al., 2014). While polar orbiting satellite observations have
- by definition - good spatio-temporal coverage in the Arctic, different factors make measurements of water vapor rather
challenging there. Techniques relying on solar radiation, e.g. the near-infrared product of the Moderate Resolution Imaging
Spectroradiometer (MODIS), are not available during polar night and need reflective surfaces such as sun glint over oceans.
The occurrence of clouds hinders water vapor measurements both for solar and thermal infrared techniques. However, even
under clear sky conditions the retrieval is difficult due to the low water vapor amounts and complex, mixed ice, snow and water
surface conditions especially in the marginal sea ice zone which is also affecting passive microwave measurements.

With their capability to gain information on water vapor under cloudy conditions, low frequency microwave imager measure-
ments now available since more than forty years have been fundamental to establish long-term climatologies of the vertically
integrated water vapor (IWV) over the ice free oceans (Mears et al., 2018; Schröder et al., 2016). To overcome the issue of the
highly variable surface emissivity in the polar regions, IWV retrieval from higher microwave frequencies, e.g the Microwave
Humidity Sounder (MHS) at millimeter wavelengths, routinely measured by microwave sounders have been proposed (Perro
et al., 2016; Scarlat et al., 2018; Triana-Gómez et al., 2020). With the launch of infrared spectrometers including several
thousands of channels, i.e. the Atmospheric Infrared Sounder (AIRS) and the Infrared Atmospheric Sounding Interferometer
(IASI), enhanced profiling capabilities for water vapor have been added and approaches for the combination with microwave
measurements are used to mitigate cloud effects. Especially AIRS has been used to study Arctic water vapor including humidity
inversions (Devasthale et al., 2016).

In order to quantify the state of the art in water vapor products being constructed for climate applications, the Global Energy
and Water Exchanges (GEWEX) Water Vapor Assessment (G-VAP) was initiated in 2011. In the framework of G-VAP, an
archive of long-term data records of water vapor related essential climate variables (including IWV) were compiled from
satellites and reanalyses (Schröder et al., 2018). When looking at IWV variability around the globe, Schröder et al. (2016)
found the highest relative standard deviation (STD) between long-term data sets in the polar regions. Thus, it is no surprise that
still strong discrepancies about pattern and magnitude of water vapor trends in the Arctic exist (Rinke et al., 2019).

This study contributes to the second phase of G-VAP by thoroughly investigating the quality of satellite and reanalysis
IWV products in the Arctic by making use of the ACLOUD/PASCAL campaign (Wendisch et al., 2019) performed in the
surroundings of Svalbard including over sea ice in May/June 2017. This period is well suited as it marks the transition period



between cold air masses with low IWV to the summer state by warm and moist intrusions from mid-latitudes. The advent
of three atmospheric rivers (ARs) during ACLOUD/PASCAL provides an interesting opportunity to investigate the impacts
of high spatio-temporal water vapor variability. ACLOUD/PASCAL also provided enhanced radiosonde measurements from
Ny-Ålesund, Svalbard, and connection to a sea ice camp at the research ice breaker Polarstern at about 82° N for evaluation of
satellite and reanalysis products.

In the past, few water vapor intercomparison studies have been performed which mainly addressed a limited set of sites
and products (Alraddawi et al., 2018; Pałm et al., 2010; Perro et al., 2016; Weaver et al., 2017). Here we aim to address
IWV performance for five frequently used global reanalyses including ERA5, the latest climate reanalysis produced by the
European Centre for Medium-Range Weather Forecasts (ECMWF), and five satellite products (Sect. 2). In this exercise, we
assess water vapor uncertainty from instantaneous (Level 2) to monthly (Level 3) products for the Arctic North Atlantic sector
which have different spatio-temporal characteristics (Sect. 3). First, we use reference IWV data from radio soundings and
continuous ground-based measurements by Global Navigation Satellite System (GNSS) and microwave radiometers (MWR)
to evaluate the quality of water vapor products on an instantaneous and local scale (Sect. 4.1). Secondly, we move to the
spatial distribution and analyze how realistic the spatio-temporal distribution is described by the different data sets on a daily
base (Sect. 4.2). Finally, we aim to connect to climate applications by investigating how uncertainties stemming from the
individual measurements, instrument limitations and sampling patterns transfer to the monthly scale (Sect. 4.3). The assessment
is concluded with an discussion and outlook to future work (Sect. 5).

## 2 Data

The ACLOUD and PASCAL campaigns (Wendisch et al., 2019) concentrated their observational efforts on Svalbard and the
Fram Strait in May and June 2017. Most important for our study are the measurements on-board the R/V *Polarstern* frozen into
the ice (around 82° N, 10° E) and the Alfred Wegener Institute for Polar and Marine Research and the French Polar Institute
Paul Emile Victor (AWIPEV) research station at Ny-Ålesund (78.92° N, 11.92° E, 2 m ASL). At both locations, frequent
radiosondes were launched and continuous IWV observations by MWR were performed. In addition, sensor synergy provides
detailed cloud characteristics (Nomokonova et al., 2019).

To broaden the scope of our study, we enlarge our study area to the North Atlantic sector (40° W to 60° E) of the Arctic,
defined as north of 60° N (Fig. 1). This area is particularly challenging for satellite retrievals as it includes land, ocean and sea
ice surfaces. Therefore, also assimilation of satellite radiances is limited and as few stations launch radiosondes on a regular
basis (Fig. 1), reanalyses are strongly depending on the underlying model (Lindsay et al., 2014). Compared to the long-term
climatology from ERA-Interim (Dee et al., 2011), both May and June 2017 were slightly drier when averaging over the full
region though some areas with moister conditions are evident, e.g. northern Russia in May. For the reference sites at R/V
*Polarstern* and Ny-Ålesund, conditions were close to the long-term mean. The 17 radiosonde stations available in the area
include both below and above normal conditions. The long-term record (Fig. 1) also indicates the strong inter-annual IWV
variability even when averaged over such a large area making the detection of trends challenging. While in the last four years





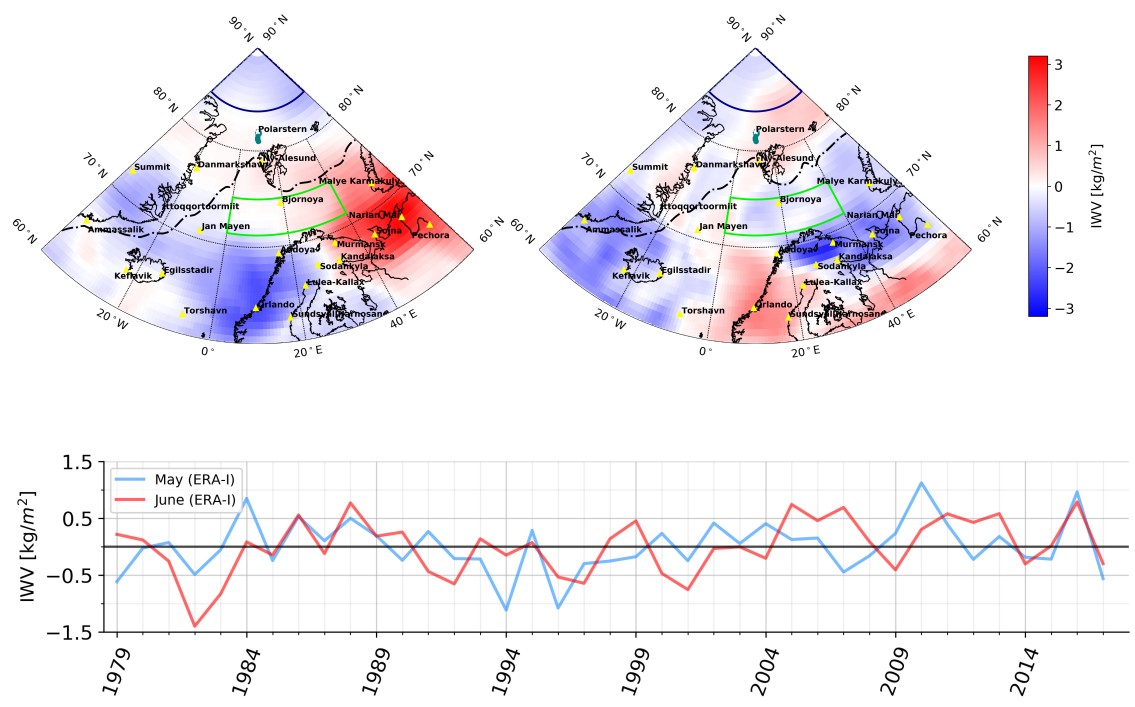

**Figure 1.** Top: Study area and location of reference stations together with map of IWV anomaly with respect to ERA-Interim long-term climatology (1979-2016) for May (left) and June (right) 2017. Yellow triangles show radiosonde stations. Average sea ice margin is given in dashed black. Two areas studied in detail are indicated in dark blue (Central Arctic) and dark green (Open Ocean); Bottom: Time series of IWV anomaly averaged over study area ($40^\circ$ W - $60^\circ$ E; $60$ - $90^\circ$ N) for May and June from 1979 to 2017.

anomalies for May and June were in phase, this has not always been the case in the past. For the spatial comparison of the different satellite and reanalysis products (Table 1), we only show the results for June 2017 while the ones for May 2017 are provided in the supplement.

## 2.1 Satellite Products

In total, six satellite products available from polar orbiting satellites operating in different parts of the electromagnetic spectrum are evaluated. Purely microwave information is used by the Advanced Microwave Scanning Radiometer2 (AMSR2) and the Microwave Integrated Retrieval System (MIRS). The infrared spectrometer IASI product also incorporates information by microwave sounders. For AIRS, a combined MW/IR product is also available. However, here we use the AIRS-only product. 95 The Global Ozone Monitoring Experiment 2 (GOME-2) makes use of spectral solar reflectance measurements. For these five satellite products orbital data are aggregated to daily and monthly means. MODIS which uses near-infrared reflectances





**Table 1.** Overview of water vapor products used in this study and their nominal resolution. Note that for cross-track imagers (e.g. IASI, MHS) the spatial resolution is highest for nadir and decreases with scan angle.

| Instrument | Platform | Product resolution | Comments | Reference |
|---|---|---|---|---|
| AIRS | AQUA | 45 km | cloud clearing at high res., purely AIRS | Aumann et al. (2003) |
| AMSR-2 | GCOM-W1 | ~20 km | all sky, ERA-Interim as a priori | Scarlat et al. (2017) |
| GOME-2 | Metop-A,B | 40, 80×40 km | no external data in retrieval | Noël et al. (2008) |
| IASI | Metop-A,B | 12 km (nadir) | combined with AVHRR and MHS | August et al. (2012) |
| MIRS | Metop-A,B, | 16 km (nadir) | variational algorithm, no NWP forecast involved | Boukabara et al. (2011) |
|  | NOAA-18,19 |  | same core software for all satellites |  |
| MODIS | AQUA, TERRA | 1 km | only daytime over reflective surfaces | Gao and Kaufman (2003) |
| Reanalysis | Producer | Original Resolution | Assimilation | Reference |
| CFSR | NCEP | ~38 km | AIRS, limited AMSU-B/MHS, IASI | Saha et al. (2014) |
| ERA5 | ECMWF | ~30 km | all sky microwave radiances | Hersbach et al. (2020) |
| ERA-Interim | ECMWF | ~79 km | AMSU-B/MHS, SSM/I, SSMIS | Dee et al. (2011) |
| JRA-55 | JMA | 1.25 x 1.25° | AMSR-2, AMSU-B/MHS, SSM/I, SSMIS | Kobayashi et al. (2015) |
| MERRA2 | NASA | ~55 km | AMSU-B/MHS | Gelaro et al. (2017) |

provides valid retrievals with much lower sampling than the others and therefore only its monthly mean IWV product is shown for completeness. An overview of the products is given in Table 1.

### 2.1.1 AIRS

Launched in May 2002 on-board of the AQUA satellite, AIRS (Aumann et al., 2003) measures radiation emitted from the atmosphere and Earth's surface in 2378 wavelength channels between 3.74 and 15.4 μm. The cross-track scanning instrument has a spatial resolution of 13.5 km in nadir decreasing to 31.5 km on the edges of the 1650 km broad swath. In this paper, the AIRS Version 6 Level 2 Standard Product (AIRS2RET) for orbital data with a 45×45 km horizontal resolution is used.

The AIRS water vapor profile product is based on a physical retrieval algorithm using AIRS IR radiances only and no MW
information. One of the first steps is to apply a cloud-clearing to the measured AIRS radiances. The retrievals of geophysical parameters are performed sequentially using the clear column radiances and an initial state being derived from a neural network approach (Susskind et al., 2014). Each geophysical parameter retrieval uses its own set of AIRS channels: for the water vapor profile retrieval, 41 channels in the spectral ranges from 1310 to 1605 $cm^{-1}$ and 2608 to 2656 $cm^{-1}$ are taken into account. IWV is directly provided in the operational product (*totH2OStd*) and has been calculated by integrating over the retrieved
specific humidity reported at 14 atmospheric layers between 1100 and 50 mbar.



An empirical error estimate is operationally provided (*totH2OStdErr*) which is calculated from a number of predictors (for details see Susskind et al., 2014). It depends strongly on the underlying surface and the presence of hydrometeors. Over cloud free ocean, uncertainty values are around 2 $\mathrm{kgm}^{-2}$ or even lower while they can reach more than 5 $\mathrm{kgm}^{-2}$ in precipitating regions. By comparing AIRS IWV to GNSS measurements, Roman et al. (2016) found a fractional error of more than 10 %

for AIRS IWV in the Arctic. Only measurements with the quality flag (*totH2OStd_QC*) Q = 0 ('highest quality') and Q = 1 ('good quality') are used in the following.

### 2.1.2 AMSR

The Advanced Microwave Scanning Radiometer 2 (AMSR2) is the successor of the AMSR and AMSR-E instruments and is operated since May 2012 on the GCOM-W1 satellite from the Japan Aerospace Exploration Agency (JAXA). The low

frequency imager has a conical scan geometry with an incidence angle of 55°. The instrument measures microwave emissions from the Earth's surface and atmosphere in 14 channels at seven different frequencies (6.9, 7.3, 10.65, 18.7, 23.8, 36.5 and 89 GHz) in vertical and horizontal polarizations (JAXA, 2016). The AMSR2 Level L1R data set (JAXA, 2013) used contains spatially consistent microwave brightness temperature observations resampled to the respective footprint sizes of the 6.9, 10.65, 23.8 and 36.5 GHz channels using the Backus-Gilbert method (Backus and Gilbert, 1968).

Integrated water vapor is acquired by an optimal estimation method (OEM) (Scarlat et al., 2017). It retrieves ensembles of surface and atmospheric parameters in the Arctic and it can use input from all AMSR2 channels. For this study a special configuration of the OEM was implemented which uses all channels between 18.7 and 89 GHz resampled to the footprint of the 23.8 GHz channels. This input combination was chosen because it provides a better resolution/sensitivity ratio than using the full AMSR2 channel suite. The method inverts the Wentz radiative transfer forward model (Wentz and Meissner, 2000)

to find a set of geophysical parameters that best fit the measured satellite top of atmosphere (TOA) brightness temperatures. Seven geophysical parameters, i.e. integrated water vapor, liquid water path, wind speed, sea surface temperature, ice surface temperature, total ice concentration and multiyear ice fraction are retrieved simultaneously by OEM. The retrieval results are of the same spatial resolution as the lowest frequency channel involved, i.e. 20 km.

Surface emissivity is needed to initialize the forward model and implement the atmospheric correction. For open ocean, the

surface emissivity is simulated by the forward model using physical temperature, salinity and surface roughness. For sea ice, the surface emissivity is a linear combination of ice types areal fraction and channel specific empirical monthly emissivities from Mathew et al. (2009). For water vapor, the 23.8 GHz water vapor absorption channels and the 89 GHz show highest sensitivities and information content. Uncertainties for IWV are at a 2 to 3 $\mathrm{kgm}^{-2}$ level depending on the ice concentration (Scarlat et al., 2017, 2020). Hereafter this product is called AMSR.

### 2.1.3 GOME-2

The Global Ozone Monitoring Experiment 2 (GOME-2) is a grating spectrometer covering the spectral range between about 240 and 780 nm (Munro et al., 2016). It is part of the payload of the series of Meteorological Operational (Metop) satellites with Metop-A (launched October 2006) and Metop-B (launched September 2012) in orbit during the time period of





ACLOUD/PASCAL. The spatial resolution of the used GOME-2 measurements is $40 \times 40$ km for Metop-A and $80 \times 40$ km for

Metop-B with a swath width of 960 and 1920 km, respectively.

The GOME-2 total column water vapor (TCVW here called IWV) data have been derived with the Air Mass Corrected Differential Optical Absorption Spectroscopy (AMC-DOAS) algorithm (Noël et al., 2008, and references therein). The AMC-DOAS product is defined as the total column water vapor w.r.t. mean sea level, so it will be typically too high for high surface elevation (which is the case for Greenland). The AMC-DOAS method is applied to sun-normalized earthshine radiance spectra

in the range between 688 and 700 nm where both water vapor and molecular oxygen ($O_2$) absorb. Only data for solar zenith angles less than 88° are used which is no problem in this season, i.e. polar day. Like in standard DOAS methods, the total amount of $H_2O$ is in principle derived from the depths of the observed differential absorption features. In addition, AMC-DOAS also (i) accounts for non-linearity (saturation effects) resulting from the strong and highly variable spectral structures of water vapor which are not resolved by GOME-2, and (ii) performs a correction for the observed light path (air mass correction)

of the retrieved water vapor total columns using $O_2$ spectral structures. The air mass correction factor is also used as an a posteriori quality check, i.e. retrieved data which require a too large correction are filtered out. This also removes most of the cloudy scenes, but an influence of remnant clouds shielding part of the water vapor columns may still be present. This may result in AMC-DOAS water vapor columns which are sometimes slightly too low.

The AMC-DOAS method products do not rely on external data (e.g. actual meteorological fields or cloud information from

other sensors or products) and therefore provide a completely independent data set. However, not making use of e.g. available a priori information also limits the accuracy of the products. In this study, we use GOME-2 AMC-DOAS water vapor data V0.5.5 with the recommended filters (maximum solar zenith angle 88°, minimum air mass correction factor 0.8) applied. The precision of the AMC-DOAS GOME-2 products (estimated from the fit residuals) is usually better than 0.5 $\mathrm{kgm}^{-2}$ at high latitudes / polar regions; however, systematic errors (especially due to non-filtered out clouds and currently unconsidered

surface elevation) may in general reach up to 5 $\mathrm{kgm}^{-2}$, but these are considered to be somewhat smaller for the conditions of the present study (low IWV, mostly ocean).

### 2.1.4 IASI

The Infrared Atmospheric Sounding Interferometer (IASI) (Blumstein et al., 2004) is a hyperspectral sounder operating in the thermal infrared. It measures between 645 and 2700 $cm^{-1}$ with a spectral resolution of 0.5 $cm^{-1}$. The observations are

acquired in a step-and-stare mode across the satellite track. The swath is approximately 2200 km wide. Each field of regard is composed of 2x2 Instantaneous Fields Of View (IFOV) within a 50 km x 50 km box. The IFOV footprints are circular with a diameter of 12 km at Nadir. They grow elliptical and grow up to 40 km in the major axis at swath edge. Like GOME-2, IASI flies on-board the Metop satellites in a sun-synchronous orbit on the 9:30AM descending node. At mid and lower latitudes, IASI revisits the same location twice per day. More frequent overpasses at high latitudes are made possible because of the

Polar orbit.

IASI flies with two microwave companions, the Advanced Microwave Sounding Unit (AMSU) and the Microwave Humidity Sounder (MHS) (Klaes et al., 2007). MHS is a cross-track sounder incorporating higher microwave channels, i.e. 89, 157, 190.3,





183.3±3.0 and 183.3±1.0 GHz. Temperature and humidity profiles belong to the suite of geophysical parameters retrieved and disseminated in near-real time by the EUropean organization for the exploitation of METeorological SATellites (EUMETSAT)
central facility (August et al., 2012). The retrieval is independent from numerical weather forecasts and solely relies on the observations. It is performed in two steps, with first a statistical, trained with machine learning approach and real observations, followed by an optimal estimation retrieval scheme in cloud-free pixels. The statistical retrieval is operative in nearly all-sky while the optimal estimation is only invoked in cloud-free pixels to refine temperature and humidity profiles further. The cloud mask is inferred from the IASI observations, supported by the collocated scenes analysis with the companion imager
instrument, the Advanced Very High Resolution Radiometer (AVHRR). Since the version 6 of the IASI L2 processor operated at EUMETSAT, the first step all-sky retrieval exploits the observations from IASI, AMSU and MHS in synergy. The total column water-vapour is integrated from the retrieved profiles and has been subject to dedicated validation against ground-based GPS IWV measurements (Roman et al., 2016). The utilisation of microwave measurements in addition to IASI enables accurate retrievals in most cloudy conditions, where clouds otherwise prevent accurate sounding down to the surface with
infrared-only retrievals.

### 2.1.5   MIRS

The Microwave Integrated Retrieval System (MIRS) IWV product from the National Oceanic and Atmospheric Administration (NOAA) is derived for different microwave satellite instruments during all weather conditions and over all surfaces in near real-time. A fast 1D-Var algorithm is used for the retrieval in which the first guess is a multi-linear regression algorithm, developed
by collocating satellite measurements with numerical weather prediction (NWP) analyses (Boukabara et al., 2011). The MIRS IWV product has a resolution of 16 km at nadir and a swath width of about 2000 km.

MIRS provides retrievals from several different satellites. Here, only retrievals from sounding instruments AMSU/MHS on-board Metop-A, Metop-B, NOAA-18 and 19 are used. We chose to omit the Global Precipitation Measurement Microwave Imager as it does not cover the central Arctic and would only provide information below 65° N. Furthermore, during the end
of June 2017, retrievals from the F17 and F18 satellites showed a sudden drop in performance and were excluded from the analysis as well By analysing microwave imager-based wind products over ocean, Robertson et al. (2020) also observed quality issues related to recent observations by SSMIS and concluded that the calibration of recent SSMIS observations needs to be carefully assessed. We only use MIRS retrievals with a quality flag of 1 (*mirs_good*). The data are checked to avoid duplicates which exist due to the overlap of orbits in the individual files. We calculate the daily means from the orbital data.

### 2.1.6   MODIS

The Moderate Resolution Imaging Spectroradiometer (MODIS) provides daytime IWV based on near-infrared (NIR) mea-surements (Gao and Kaufman, 2003). The IWV is retrieved by using the ratio of NIR water vapor absorbing channels and atmospheric window channels. From this water vapor transmittance the IWV is derived with an accuracy of 5-10 %, making use of theoretical radiative transfer calculations and a look-up-table procedure. IWV collection 6 products are available for the
MODIS instruments onboard the afternoon and morning polar-orbiting satellites Aqua and Terra separately. Combined, they

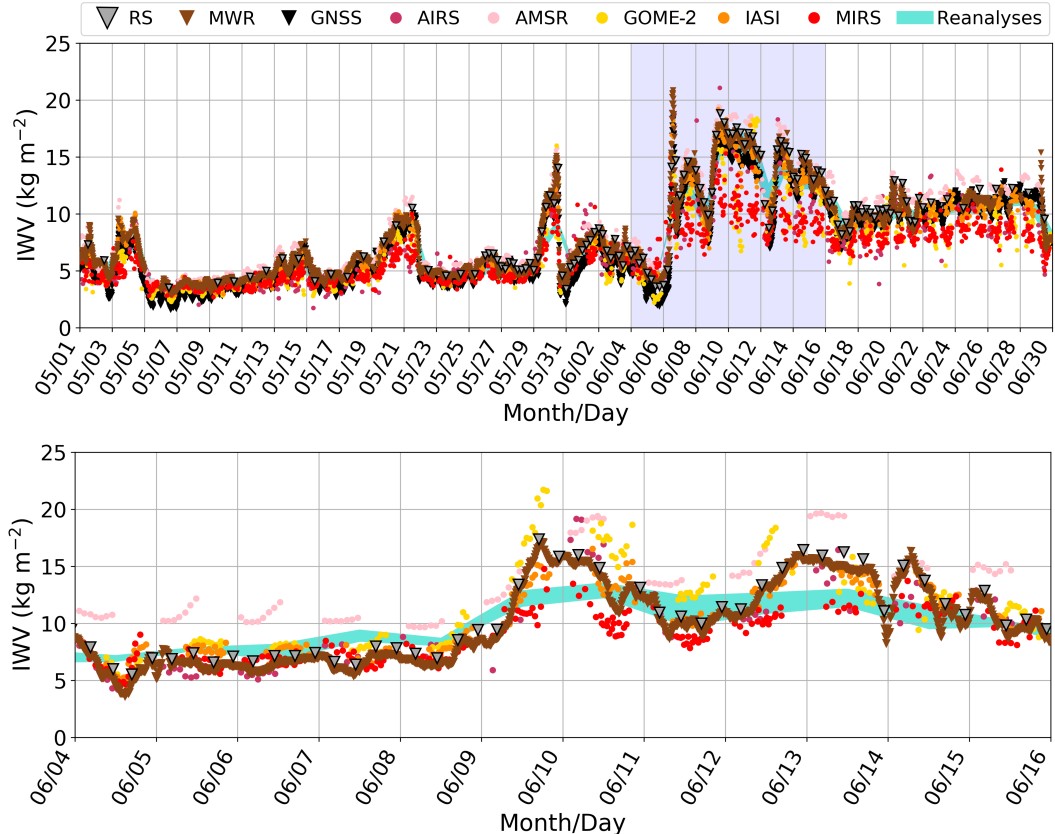

**Figure 2.** Time series of reference data (GNSS, MWR, RS), reanalyses and satellites (see legend for explanation of symbols) at the two ACLOUD central sites Ny-Ålesund (top) and the R/V *Polarstern* ice camp (bottom). The time period of the ice camp is indicated in blue in the time series for Ny-Ålesund. The cyan shaded area indicates the minimum and maximum values of the reanalyses.

provide a near global coverage twice a day. We make use of the level 3 monthly mean products, MYD08_M3 (AQUA) and MOD08_M3 (TERRA). The spatial resolution of MODIS NIR IWV products is 1 km at nadir for the orbital files and 1° for the monthly means.

Level 2 orbital data (MYD05_L2 (AQUA) and MOD05_L2 (TERRA)) will be exemplarily shown for one day highlighting the low data availability of MODIS IWV is in many parts of the Arctic regime. This is due to the inability of MODIS to penetrate clouds, which have a high occurrence over the Arctic and subarctic ocean ((Mioche et al., 2015)) and the need for highly reflective surfaces. In case of a cloudy regime only the IWV above the cloudy layer(s) can be retrieved. Therefore, daily means of IWV by MODIS are not used and only the monthly means are shown in our investigations.





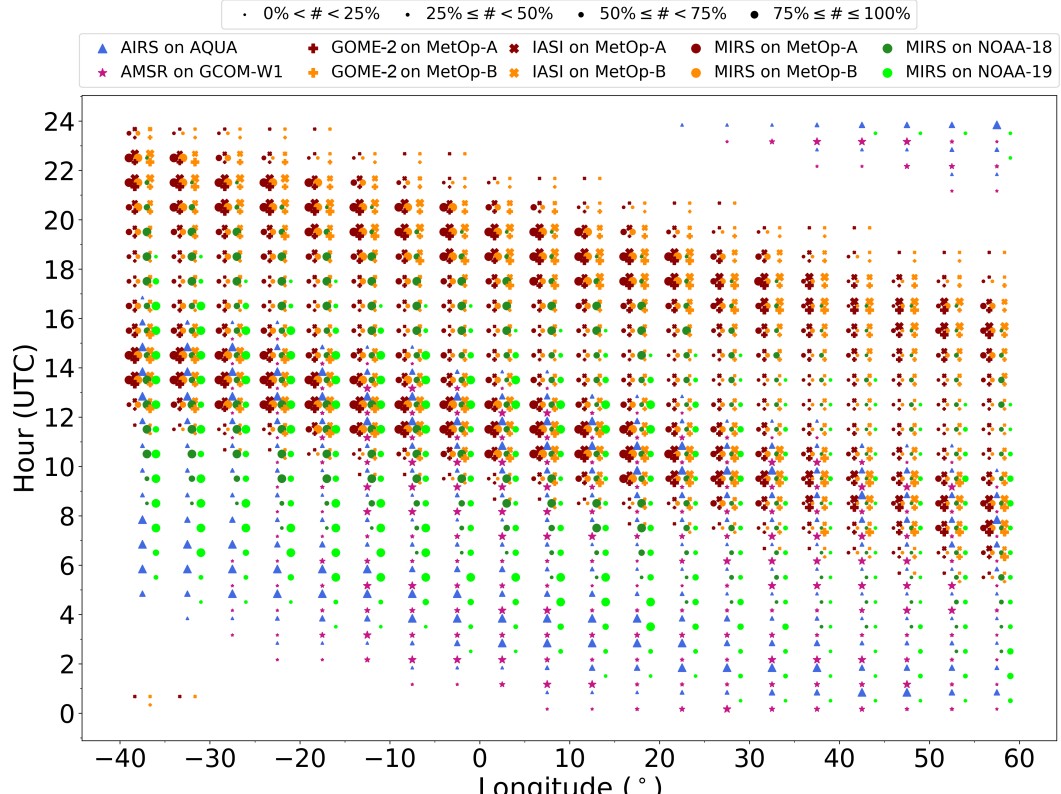

**Figure 3.** Overview on satellite data sampling for a latitude band (70-80° N) as a function of time of day and longitude (5° resolution). For each bin (1 hour, 5°) the total amount of measurements over the two month period (May-June 2017) is indicated per instrument by the size of the corresponding symbol in quantiles. The maximum number of samples per bin is about 2000 for AIRS on AQUA, 8000 for each IASI platform and about 10.000 for the MIRS products on each satellite.

## 2.2 Reanalyses

The same four modern global atmospheric reanalyses as in Rinke et al. (2019) are used in this study, i.e. the National Centers for Environmental Prediction (NCEP) Climate Forecast System Reanalysis (CFSR; Saha et al. (2014)), the European Centre for Medium-Range Weather Forecasts (ECMWF) interim reanalysis (ERA-Interim, hereafter ERAI; Dee et al. (2011), the Japanese Meteorological Agency (JMA) 55-year Reanalysis (JRA-55; Kobayashi et al. (2015)), and the NASA Global Modeling and Assimilation Office (GMAO) Modern-Era Retrospective Analysis for Research and Applications, version 2 (MERRA2; Gelaro 225 et al. (2017)). Due to the lack of any reference data and in order to compare with Rinke et al. (2009), the median of the four reanalyses is taken as reference.





Furthermore, we explore the performance of the next-generation ECMWF reanalysis, ERA5 (Hersbach et al., 2020), which has a higher spatial resolution as all other global reanalyses except CFSR (Table 1). For our studied period of 2017, the CFSv2 operational analysis is used, which has a similar high resolution ($\sim$27 km) as ERA5 ($\sim$30 km). Furthermore, CFSv2 involves

a coupling of atmosphere and ocean and an interactive sea-ice model.

## 2.3   Reference IWV measurements

In order to evaluate the quality of the spatial products we make use of radiosondes and ground-based remote sensing by GNSS and by MWR at selected stations (Fig. 1). Radiosondes were taken from the Integrated Global Radiosonde Archive (IGRA) (Durre et al., 2006) with the exception of the soundings from R/V *Polarstern* available at Schmithüsen (2017). For

Ny-Ålesund, the accuracy of the lower vertically resolved IGRA profiles was checked by comparing all 143 ascents with their high resolution with their high resolution data version (Maturilli, 2017a, b). counterparts yielding an excellent agreement with a root mean square deviation (RMSD) of 0.1 $\mathrm{kgm^{-2}}$. During the ACLOUD/PASCAL campaigns, RS92 (RS41) radiosondes were launched at R/V *Polarstern* (Ny-Ålesund) every 6 hours (00, 06, 12, 18 UTC synoptic times). By default, the data were transmitted to WMO's Global Telecommunication System (GTS), and were thus available for assimilation in NWP products

and atmospheric reanalyses.

At Ny-Ålesund (Svalbard), the delay of the GNSS signal between the satellite and the ground stations is used to derive IWV. Due to the use of rather low microwave frequencies, all-weather measurements can be conducted. The data were processed by the GeoForschungsZentrum Potsdam using the European Plate Observing System (EPOS) software with temporal resolution of 15 min and an accuracy of 1-2 $\mathrm{kgm^{-2}}$ (Ge et al., 2006; Gendt et al., 2004).

Continuous time series with sub-minute temporal resolution are available from MWR, i.e. the Humidity And Temperature Profiler (HATPRO; Rose et al., 2005) operated on-board of R/V Polarstern (Griesche et al., 2020) and at Ny-Ålesund (Nomokonova et al., 2019). Herein, IWV is retrieved from measurements along the 22.235 GHz water vapor absorption line by a linear regression algorithm following Löhnert and Crewell (2003). A decade long training data set of GRUAN sondes from Ny-Ålesund has been used in the regression algorithm for the AWIPEV and Polarstern measurements. HATPRO provides

IWV during all weather conditions except for cases when the radome of the instrument is wet, e.g. due to precipitation. The accuracy is estimated to be about 0.5 $\mathrm{kgm^{-2}}$.

Continuous measurements by the MWR and GNSS are able to capture the temporal variability rather well and complement the radiosondes (Fig. 2). Compared to the radiosondes, MWR (GNSS) IWV has a bias of -0.3 (-1.2) $\mathrm{kgm^{-2}}$ and a RMSD of 0.5 (1.3) $\mathrm{kgm^{-2}}$ at Ny-Ålesund. Consequently, GNSS is by 0.6 $\mathrm{kgm^{-2}}$ lower than the MWR which is also well visible in the

time series of one individual day (Fig. 4). It is worth to note that these skill scores derived for the ACLOUD period are well in line with those calculated over the period from 2015 to 2018 (not shown) indicating that the biases are rather stable. The STDs between the three different instrument types are lower than 0.8 $\mathrm{kgm^{-2}}$ and thus make them well suited for the following evaluation of the spatial products.





## 3   Matching satellite/reanalysis and reference IWV measurements

All satellites considered have sun-synchronous orbits with orbit durations of about 100 min. For 2017, GOME-2, IASI and MIRS observations are available from the Metop-A and Metop-B satellites while AMSR and AIRS are only on-board of one satellite reducing their number of samples. Figure 3 illustrates the excellent sampling of the polar orbiters at high latitudes and the complementarity of the morning (Metop) and afternoon (NOAA) orbit providing nearly continuous sampling over two thirds of the day. Nevertheless sampling strongly depends on the longitude and it becomes clear that good matching with the

synoptic launch times of radiosondes is often not possible especially for the eastern regions and the launch time at 0 UTC.

The satellite IWV products were all provided as orbital data on a pixel-basis. For the intercomparison with reference data, all pixels with valid IWV retrievals in a radius of 50 km around the individual stations (Fig. 1) were extracted. As an example, Fig. 2 shows the good temporal sampling of satellite and reference (MWR, GNSS, radiosonde) measurements over the full two month period for Ny-Ålesund and the two weeks time period for the ice embarkment of the R/V *Polarstern* close to $81°$ N.

During the shorter period it can be seen that the overpasses by Metop-A and Metop-B which host GOME-2, IASI and MHS cover the time period between roughly 8 and 18 UTC rather well for these two reference sites (cf. also Fig. 3) while AMSR measures in the first half of the day (Fig. 2). The MIRS product covers the widest range as in addition to the Metop satellites also NOAA-18 and 19 are used.

In order to compare the satellite with reference data different criteria are used: (i) the high temporally resolved ground-based

MWR data are averaged to 15 min means to match the GNSS measurements. Their temporally closest measurement to the radiosonde synop time is used for comparisons. (ii) a time window of $\pm30$ minutes with respect to the radiosonde time is used to identify corresponding satellite measurements. Note, that a larger window length of $\pm 1$ hour does not drastically enhance the number of matched samples for the radiosondes due to the fixed launch times at most stations. (iii) All IWV satellite measurements with center pixel location within a 50 $\mathrm{km}$ circle around the location of the reference site are used to calculate

mean IWV and its STD. The same exercise has been performed for a larger search radius of 100 km to check the sensitivity. (iv) To eliminate outliers only IWV values between 0 and 30 $\mathrm{kgm^{-2}}$ are used which is sensible as IWV varies between 3 and 22 $\mathrm{kgm^{-2}}$ for the Ny-Ålesund and R/V *Polarstern* sites (Fig. 2). While we are aware that most satellite retrievals work best over ocean surfaces (well characterized microwave emissivity) and are affected by differences in orography, we consciously use all conditions for our assessment as we aim at a climatological sound data set. In order to estimate the influence of orography

and surface emissivity, all measurements were classified according to their position over water or land. This is for example of interest for Ny-Ålesund which is located in a fjord surrounded by mountains.

For maps of daily and monthly IWV, the reanalysis products were interpolated to the ERA-Interim grid with $0.75°\times0.75°$ resolution. All orbital satellite data for a day is assigned to the same $0.75°\times0.75°$ latitude-longitude grid spanning the study area. The daily means are calculated as the arithmetic mean per grid cell, using only measurements which fulfill the quality

criteria (Sect. 2.1). Due to the different satellite orbits, sampling differs for the different products (see above). Note that due to the meridian convergence this means that at $70°$ N, the resolution along the latitude circle is only 24 km while it is 83 $\mathrm{km}$ in meridional direction.



**Figure 4.** Illustration of the AR event on 6 June 2017 as provided by four different reanalyses at 12 UTC and instantaneous satellite measurements (closest orbit in time). The magenta line depicts the region where IWV is higher than an IWV threshold based on the saturated IWV and an AR coefficient (Gorodetskaya et al., 2014) using ERA-Interim reanalysis. For the remaining reanalysis data sets, the line was interpolated from ERA-Interim. Central Arctic (dark blue) and Open Ocean (green) regions are marked. Bottom: Time series at Ny-Ålesund for 6 June 0 UTC to 7 June 0 UTC from reference data, reanalyses and satellite measurements within 50 km of the site.



## 4  Results

### 4.1  Direct comparisons of satellite/reanalysis with reference data

The ACLOUD/PASCAL campaign offers a wide range of IWV conditions to investigate the performance of IWV products. The time series at Ny-Ålesund (Fig. 2) shows first an unusual dry (and cold) phase followed by an unusual wet (and warm) period (30 May - 12 June) connected with high IWV variability as already described by Knudsen et al. (2018). Afterwards normal conditions prevailed. Further it reveals several IWV peaks (Fig. 2) from which three could be identified as ARs following the definition by Gorodetskaya et al. (2014). Here we choose the AR event from 6 June 2017 at 12 UTC to illustrate the capabilities

of the different products (Fig. 4). By definition reanalyses provide information across the full region revealing the maximum IWV of about 25 $\mathrm{kgm}^{-2}$ west of Novaya Zemlja from where an elongated band of IWV stretches westward passing Svalbard and dissolves north of Iceland with extended cloudiness and convective precipitation. As the data are shown here in their original resolution, differences between reanalyses with respect to gradients, coastal features and maximum IWV are evident showing the better representation of small scale features in the high resolution ERA5 reanalysis.

The temporally closest Metop satellite overpass providing GOME, IASI and MIRS products is on the descending branch of the orbit while AQUA (AIRS) and GCOM-W1 (AMSR) are on a ascending one (Fig. 4). The individual orbits of the satellite products demonstrate the different swath widths as well as the limitations of the products. The AIRS only product shows the largest spatial gaps due to the limitation of infrared measurements in the presence of clouds and precipitation. AMSR low frequency microwave information is also available in cloudy regions but due to the complex emissivity over land only

measurements over ocean and sea ice are provided. GOME-2 provides retrievals over all surfaces but cloud disturbances lead to data gaps close to Svalbard and north of Iceland. For the latter region MIRS, which retrieves several parameters simultaneously, indicates precipitation. Note, that due to the dominance of the precipitation signal the information on water vapor can be obscured for heavy rain events.IASI and MIRS, which mitigate cloud influence by the use of microwave radiances in their retrieval schemes, have nearly complete coverage. As already mentioned, MODIS only provides rather limited information as

it can derive IWV only under cloud free conditions over strongly reflective surfaces.

The daily time series of the MWR at Ny-Ålesund (Fig. 4, bottom) shows that IWV rapidly increases by about 15 $\mathrm{kgm}^{-2}$ within only 5 hours reaching its peak value of 22 $\mathrm{kgm}^{-2}$ around 14 UTC before declining again with similar speed but arriving at a higher level (10 $\mathrm{kgm}^{-2}$). The ground-based MWR agrees well with the radiosondes launched during that day with the exception of 18 UTC which could be caused by the drift of the radiosonde across the strong IWV gradient. GNSS and MWR

have a slight mismatch in the diurnal cycle that might be due to the slant path between GNSS satellites and the ground receiver or problems in the derivation of the mean weighted temperature used in the GNSS retrieval (Morland et al., 2009). Generally, it becomes clear that dense temporal and spatial sampling with high resolution is necessary to characterize such an event.

The time series during the AR event (Fig. 4, bottom) reveals differences of up to 6 $\mathrm{kgm}^{-2}$ at 12 UTC between MERRA2 and JRA-55 which is likely due to mismatches in the movement of the AR. Looking at the satellite products shows an even larger

spread among the measurements: AMSR which has eight overpasses over Ny-Ålesund between 01 and 13 UTC agrees very well with the ground-based MWR before the arrival of the AR. During this time also little variability between pixels within

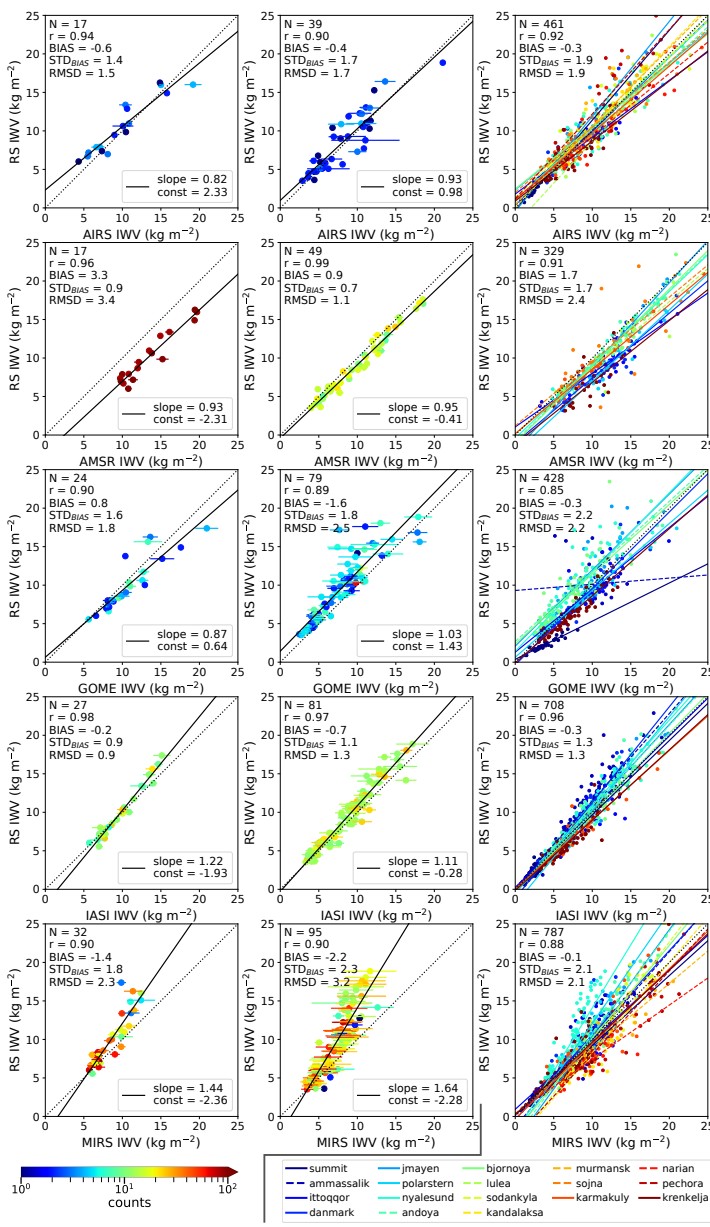

**Figure 5.** Scatter plots for radiosondes launched at Polarstern (left column), Ny-Ålesund (middle column) and for all other radiosonde stations (right column) with corresponding (± 30 min and 50 km radius) satellite measurements (from top to bottom row: AIRS, AMSR, GOME-2, IASI, MIRS) above all surfaces. All satellite pixels falling into this criterion have been averaged and their STD is indicated by the width of the line. The number of averaged pixels is indicated by color for the first two columns only.





the 50 km radius is observed which increases strongly with the arrival of the AR. IASI provides the best agreement for this case while GOME-2 and MIRS strongly underestimate the AR maximum. In fact, MIRS seems to have difficulties in retrieving higher IWV values at all as also indicated in the moist June period (Fig. 2). Consistently with the orbit characteristics of the satellites (Fig. 3), towards the end of the day no satellite matches can be found for Ny-Ålesund.

To quantitatively assess the accuracy of the IWV products, in a first step pairs between radiosonde measurements and corresponding products which fulfill the matching criteria (Sect. 3) are compiled (Fig. 5) and skill scores, i.e. bias, correlation coefficient (r), RMSD and STD (i.e. the bias corrected RMSD), are computed. For R/V *Polarstern* (Fig. 5, left column) only few matched samples (between 17 to 32) are available due to the limited deployment time. IASI retrievals can clearly be identified showing the best performance with the lowest bias (-0.2 $\text{kgm}^{-2}$), highest correlation (0.98) and lowest STD (0.9 $\text{kgm}^{-2}$). Each individual radiosonde match is an average of about ten individual pixels and their low variation indicates relatively homogeneous conditions around the site. This is also seen by AMSR which has as well a rather low STD (0.9 $\text{kgm}^{-2}$) but is affected by a strong bias (3.2 $\text{kgm}^{-2}$).

At Ny-Ålesund (Fig. 5, middle column) where more than twice as many matches are available, AMSR shows the highest correlation (0.99) and lowest STD (0.7 $\text{kgm}^{-2}$) followed by IASI (r=0.97, STD=1.1 $\text{kgm}^{-2}$).Interestingly, the bias of AMSR is strongly reduced (0.88 $\text{kgm}^{-2}$) compared to the ice floe region at R/V *Polarstern* indicating an emissivity issue for sea ice. IASI shows a negative bias of -0.7 $\text{kgm}^{-2}$ which can be explained by the orography around the launch site as expressed by the reduction of the bias to -0.1 $\text{kgm}^{-2}$ when only pixels above water are considered. Note, that no distinct changes in the other scores occur. Surprisingly, although rather similar to IASI, AIRS shows degraded performance for both sites with STD being about 0.5 $\text{kgm}^{-2}$ higher than for IASI. This indicates the benefit of the different IASI retrieval strategy, e.g. individual pixels, inclusion of microwave information.

The performance of GOME-2 substantially degrades for IWV values above 10 $\text{kgm}^{-2}$ leading to much higher STDs at R/V *Polarstern* (1.6 $\text{kgm}^{-2}$) and Ny-Ålesund (1.8 $\text{kgm}^{-2}$) than shown by AIRS, AMSR and IASI. Considering only water surfaces even slightly worsens the scores (not shown). GOME-2 and MIRS show a similar correlation (0.90) for both sites. MIRS which has the highest number of matches per individual radiosonde reveals a strong underestimation for IWV values higher than about 10 $\text{kgm}^{-2}$ both for R/V *Polarstern* as well as for Ny-Ålesund where nearly 100 radiosondes are compared. This results in a slope in the regression of 1.44 (1.64) for R/V *Polarstern* (Ny-Ålesund) which leads to a much narrower retrieved IWV frequency distribution than measured by radiosondes thus underestimating IWV variability. For MIRS, the scores improve when only water surfaces and a larger search radius (100 km) is used, i.e. correlation increases from 0.90 to 0.95 and STD reduces from 2.3 to 1.6 $\text{kgm}^{-2}$ (not shown).

When considering all other radiosonde stations together (Fig. 5, right column; Table A1; for locations of stations see Fig. 1), one has to take caution because the different orbit characteristics together with the fixed launch times produce different sets of matched stations for the different products (Fig. 3). Because MIRS makes use of four different satellites most samples (914) are found. However, MIRS is clearly the product revealing the strongest scatter. This not necessarily due to the quality of the satellite product but might arise from the consideration of different samples. NOAA-19 has several matches with Russian RS stations while these are rare for Metop satellites. For Russian RS stations a slope parameter lower than one indicates an



overestimation of MIRS which could be due to problems with surface emissivity but an underestimation of IWV by the RS can also not be excluded. On the other hand, RS stations in Greenland and North Scandinavia mainly show slopes larger than 1 indicating an underestimation of MIRS. The use of different radiosonde sensors in different countries or regions is known to result in an uneven distribution of temperature and humidity biases across geopolitical borders (Soden and Lanzante, 1996; Ho et al., 2017; Ingleby, 2017). With a correlation of 0.96, STD of 1.29 $\mathrm{kgm}^{-2}$ and RMSD of 2.21 $\mathrm{kgm}^{-2}$ IASI again shows the best performance of all satellite products. While the scatter is much lower than in the case of MIRS the same trend in respect to over/underestimation for different stations can be seen. Detailed statistics separately for all individual radiosonde stations are given in the appendix (A1). Note, that a direct comparison between the radiosonde measurements and reanalyses has not been pursued as the radiosondes are assimilated into reanalyses.

## 4.2 Assessment of daily mean data

With the uncertainty of the individual satellite measurements addressed by the direct intercomparison we now aim to investigate the suitability of the satellite products for climate studies. To better understand how uncertainties are transferred we first compile daily mean values from orbital data which are then aggregated to monthly means (cf. Sect. 3). Assessing the quality of these products with reference measurements is only possible at sites with continuous ground-based measurements reducing the data set notably. Therefore, in order to better identify the differences between the products we look at anomalies in respect to the median of the four classical reanalysis (CFSR, ERA-Interim, JRA55, MERRA2). In case of random noise, a distinct reduction in uncertainty due to averaging should occur while systematic errors should become more pronounced. In addition, the irregular sampling of satellite data can introduce errors which will depend on the prevailing weather conditions.

The AR event of 6 June 2017 with high IWV contrasts is used to study the differences between IWV products now on a daily mean basis. Compared to the snapshot at 12 UTC (Fig. 4), the AR is smoothed in the reanalyses median over the full day but still visible (Fig. 6). Differences between reanalyses are around 10% with higher deviation along coastlines and strong orography that can easily be explained by differences in their original resolution. In that sense, it is not a surprise that CFSR with its higher resolution shows even higher deviation in some of these areas, e.g. coast of Greenland, as the orography is smoothed less giving lower IWV at grid points with higher altitude. Over the open ocean, the spatial structure of the differences between the classical reanalyses does not seem to be strongly related to the AR shape with the exception of the dry line close to 40° E in ERAI which might hint at differences in the data assimilation of the different reanalyses. Instead already on the daily scale some systematic differences occur over sea ice / ocean that will be discussed later on. Interestingly, the new high resolution reanalysis ERA5 (not part of the reanalyses median) substantially differs from the heritage product ERAI though some similarities such as the positive (moist) difference over sea ice appear.

In general, the deviations of the satellite products to the reanalyses median for 6 June 2017 are about a factor two higher than those of the reanalyses. Different to the reanalyses, the satellite products all have a different sampling density per grid cell due to the different orbit characteristics (Fig. 3). Due to their orbit for these polar orbiting satellites the best sampling globally occurs in a band centered around 73° N latitude. AIRS and GOME-2 have the lowest number of samples while IASI and MIRS have around 50 individual measurements per grid cell. In fact IASI and MIRS show very similar geographical structures in



**Figure 6.** Relative difference of the daily means of the reanalyses and satellite products to the reanalyses median (CFSR, ERA-Interim, JRA-55, MERRA2; lower right plot) for 6 June 2017. The green line indicates the sea ice edge from AMSR sea ice data. Central Arctic (dark blue) and Open Ocean (green) regions are marked.

their differences which is no surprise as the IASI product incorporates the microwave measurements on the Metop satellites. The strong bands of positive and negative deviations along the northward extent of the AR can reaching up to 50 % (Fig. 6) can be attributed to sampling differences due to the fast movement of the AR. As the reanalyses median is only computed from the





6 hourly IWV values the satellites are likely able to better capture this development. This is supported by the similar structures

evident in the ERA5 daily mean product (weighting all 1h time steps). The resemblance between ERA5 with IASI as well as MIRS seems to be limited to open ocean surfaces where ERA5 assimilates their data. Before we move on to a discussion on systematic effects we want to investigate the "weather related" averaging effects in more detail.

During the strong AR event (Fig. 4) clearly high deviations of several $\text{kg m}^{-2}$ between different products are possible if the daily mean is calculated from few samples - but how frequently does this occur? To investigate the limitations due to

infrequent temporal sampling we use the continuous MWR IWV at Ny-Ålesund which is available in sub-minute resolution. The basic idea is to mimic the sampling characteristics of other observation systems such as radiosonde stations or sporadic satellite overpasses. During ACLOUD/PASCAL daily mean values calculated from the four time steps such as from six-hourly radiosonde launches would give a negligible deviation on average with a STD of 0.3 $\text{kgm}^{-2}$ but individual deviations of 1 $\text{kgm}^{-2}$ or more occur. When looking at a multi-year data set (2015-2018; not shown), no bias but a STD of 0.5 $\text{kgm}^{-2}$

is present. Most of the Arctic radiosonde stations launch sondes twice or sometimes only once per day. In this case, the deviations from the true daily mean are even worse. Generally, the STD depends on IWV itself with a relative STD of 5 % with samples every 6 hours and degrades to 10% if two samples (0 and 12 UTC) are used. Taking only the 12 UTC measurement as representative for the day, only slightly worsens the situation with a relative STD of 12 %.

In the comparison of the daily mean differences, several geographic features appear that point to systematic effects (Fig. 6).

Consistent with the previously discussed time series for R/V *Polarstern* (Fig. 2), AMSR shows a positive bias over the sea ice of more than 30%. Over sea ice, also GOME-2 shows a positive bias pointing at an issue with surface reflectivity for GOME-2 and surface emissivity for AMSR. The positive bias of GOME-2 over Greenland is partly due to the definition of the product that provide the column above mean sea level. Due to the high elevation of the Greenland ice sheet the lowest absolute IWV occurs here (see reanalyses median). Therefore small absolute IWV differences lead to high relative differences (also seen

by AIRS and IASI). MIRS shows a high positive overestimation over the Russian land area consistent with the radiosonde intercomparison.

To better understand systematic features, we study two regions with relatively homogeneous surface conditions over the course of the ACLOUD/PASCAL campaign (cf. Fig. 1). The first region is the high Arctic above 84° N (in the following called "Central Arctic") where no surface reference measurements exist and biases between reanalyses are evident: While ERAI and

ERA5 show positive deviations over the full sea ice covered area, CFSR and MERRA2 show negative deviations as already noted by Rinke et al. (2019). The second area concerns the ice free North Atlantic towards the Barents Sea (0-40° E, 72-75.75° N; in the following called "Open Ocean").

Over the open ocean area, low frequency microwave observations should have the best performance due to the low and well characterized surface emissivity. Therefore it is no surprise that AMSR shows the lowest STD (0.9 $\text{kgm}^{-2}$, Fig. 7, Table 2)

compared to the reanalyses median which might also be due to its assimilation into the reanalyses. The same holds for MIRS and IASI (with STD=0.6 $\text{kgm}^{-2}$ and STD=0.7 $\text{kgm}^{-2}$, respectively) that incorporate higher microwave frequencies. With frequent low-level cloudiness over the North Atlantic, it is no surprise that the pure thermal IR (AIRS, Bias=1.2 $\text{kgm}^{-2}$, STD=1.1 $\text{kgm}^{-2}$) and solar (GOME-2, Bias= 2.8 $\text{kgm}^{-2}$, STD=1.1 $\text{kgm}^{-2}$) have difficulties also reflected in the much poorer





**Figure 7.** Joint distribution of daily means from the satellite (x-axis; AIRS, AMSR, GOME-2, IASI, MIRS) and reanalyses median (y-axis CFSR, ERA-Interim, JRA-55, MERRA2) for the $0.75° \times 0.75°$ grid points in the Central Arctic (left), Open Ocean (middle) and the full region (right). The time period is May to June 2017. The color indicates the relative fraction of the IWV.





correlation. The different behavior between the different satellite products is further illustrated by looking at the temporal
development over the two months (Fig. 8). AMSR, IASI and MIRS show overall similar performances as the reanalyses well
reproducing IWV day-to day variability with daily means between 5 and 15 $\mathrm{kgm^{-2}}$. In this homogeneous region, reanalyses
are highly consistent with STD of 0.3 to 0.4 $\mathrm{kgm^{-2}}$ (Fig. A1). However, the reanalysis bias (Table 2) varies between -0.7
$\mathrm{kgm^{-2}}$ (CFSR) and +0.6 $\mathrm{kgm^{-2}}$ (JRA55) which might be related to differences in data assimilation or model physics. In fact,
the reanalyses differ in their differences more than the three satellite products which vary between -0.4 $\mathrm{kgm^{-2}}$ (AMSR) and
+0.1 $\mathrm{kgm^{-2}}$ (IASI). Therefore one might conclude that in open ocean areas these satellite products can be used to further
improve reanalyses.

For the sea ice region, one can nicely see how the rather constant dry conditions in the central Arctic prevailing in the first
half of May are changed by moisture transport from the south (Fig. 8). This transport mostly takes place by individual events
such as the AR event discussed before that results roughly in a tripling of IWV in the ACLOUD/PASCAL period. The sporadic
nature in the transport seems to cause a larger spread between reanalyses and also satellite products (cf. period from June 10th
onward). Similar to the open ocean, relative differences between reanalyses are up to $\pm$ 10 %. Out of the satellite products only
IASI shows similar performance as reanalysis. While GOME-2 showed strong IWV underestimation over the dark open ocean,
its performance over the bright sea ice is much more satisfactory especially at the lower IWV end. This also becomes visible
when time series of area averaged daily IWV are considered (Fig. 8). As indicated before, AMSR shows a clear overestimation
over the sea ice in the central Arctic and similar high scattering than AIRS. This might also be due to the high sensitivity in
respect to the surface emissivity, e.g. leads, polynyas, to which the higher frequency products (IASI, MIRS) are less effected.
AIRS only has 17 % data coverage as it only flies on one satellite and employs a rigid cloud filtering 4 so that it is difficult to
draw solid conclusions.

For generalizing the results over the full time period (May and June 2017) and full region, all daily mean values are compared
with their counterpart from the reanalyses median (Fig. 7; Table 2). While the reanalysis data over the full regions are highly
correlated (>0.99, cf. Fig. A1) the correlations reduce for the satellite products. IASI is closest to the reanalyses with small bias
(0.2 $\mathrm{kgm^{-2}}$) and lowest STD (0.9 $\mathrm{kgm^{-2}}$) together with a high coverage of the domain (96.7 %). Interestingly, AIRS which
is a pure IR product performs worse than IASI (Bias 0.8 $\mathrm{kgm^{-2}}$, STD=1.9 $\mathrm{kgm^{-2}}$) and also shows more data gaps due to a
lower spatial coverage and likely also due to cloud filtering. The question is whether these differences are caused solely by
the incorporation of microwave measurements into the IASI product or if the incorporation of NWP background data leads the
good agreement with reanalyses.

### 4.3   Assessment of monthly mean data

For climatological analyses, monthly mean values are typically the shortest time scale considered. Therefore, we further analyse
the different monthly means for May (Fig. A2) and June (Fig. 9) 2017 - again in terms of their deviation to the reanalyses
median (anomalies). These two months are rather interesting as the monthly mean IWV increases roughly by a factor of two
from May to June indicating the transition into the Arctic summer. Nevertheless, for all products the anomalies (in respect to
the reanalyses median) are rather similar in their geographical patterns for both months. In particular, contrasts between open



**Figure 8.** Daily time series of area averaged IWV for the central Arctic (top) and open ocean (bottom) areas for different products as indicated in the legends. The relative difference is given in respect to the reanalyses median (CFSR, ERAI, JRA55, MERRA2; dashed black line). The shaded area indicates the minimum and maximum values of the reanalyses.

ocean, sea ice and Greenland are evident. AMSR and GOME-2 clearly overestimate IWV over sea ice by more than 20 %. While AMSR still has slight moist anomalies over ocean, GOME-2 underestimates here by more than 20%. A similar pattern
(overestimation above sea ice, underestimation over ocean) albeit with lower magnitude (about 10 %) can be seen for AIRS





**Table 2.** Skill scores for the intercomparison of daily IWV mean to reanalyses median in May and June 2017 for all valid data pairs in terms of bias, STD and RMSD (all in $\mathrm{kgm^{-2}}$) and correlation coefficient (r). Results have been calculated for the central Arctic, open ocean and the complete study taking into account the area of the different grid cells. N denotes the number of samples.

| Product | CFSR | ERA5 | ERAI | JRA55 | MERRA2 | AIRS | AMSR | GOME2 | IASI | MIRS |
|---|---|---|---|---|---|---|---|---|---|---|
| **Central Arctic** | | | | | | | | | | |
| N | 73566 | 73566 | 73566 | 73566 | 73566 | 12435 | 65392 | 73566 | 63002 | 60321 |
| N in % | 100 | 100 | 100 | 100 | 100 | 17.0 | 88.9 | 100 | 86.3 | 82.6 |
| Mean | 6.0 | 6.7 | 7.0 | 6.7 | 6.5 | 7.8 | 8.8 | 7.1 | 6.8 | 6.4 |
| Bias | 0.5 | -0.2 | -0.4 | -0.1 | 0.0 | -0.4 | -2.2 | -0.5 | -0.2 | 0.2 |
| RMSD | 0.8 | 0.4 | 0.5 | 0.5 | 0.4 | 1.3 | 1.4 | 1.1 | 0.9 | 0.9 |
| STD | 0.4 | 0.3 | 0.2 | 0.3 | 0.3 | 1.0 | 0.9 | 0.6 | 0.6 | 0.6 |
| r | 0.98 | 0.99 | 0.99 | 0.99 | 0.99 | 0.92 | 0.90 | 0.95 | 0.96 | 0.95 |
| **Open Ocean** | | | | | | | | | | |
| N | 16470 | 16470 | 16470 | 16470 | 16470 | 14502 | 16470 | 16465 | 16468 | 14887 |
| N in % | 100 | 100 | 100 | 100 | 100 | 88.1 | 100 | 100 | 100 | 90.4 |
| Mean | 10.3 | 9.3 | 9.3 | 9.0 | 9.8 | 8.8 | 10.1 | 6.8 | 9.5 | 9.8 |
| Bias | -0.7 | 0.3 | 0.3 | 0.6 | -0.2 | 1.2 | -0.4 | 2.8 | 0.1 | 0.0 |
| RMSD | 0.5 | 0.5 | 0.4 | 0.4 | 0.4 | 1.6 | 0.9 | 1.6 | 1.0 | 1.0 |
| STD | 0.3 | 0.4 | 0.3 | 0.3 | 0.3 | 1.1 | 0.6 | 1.1 | 0.7 | 0.6 |
| r | 0.99 | 0.99 | 0.99 | 0.99 | 0.99 | 0.88 | 0.97 | 0.89 | 0.96 | 0.96 |
| **Complete Area** | | | | | | | | | | |
| N | 324520 | 324520 | 324520 | 324520 | 324520 | 221357 | 236480 | 323729 | 313927 | 297427 |
| N in % | 100 | 100 | 100 | 100 | 100 | 68.2 | 73.1 | 99.8 | 96.7 | 91.7 |
| Mean | 9.2 | 8.9 | 9.0 | 8.6 | 9.1 | 9.8 | 11.8 | 8.2 | 10.2 | 10.4 |
| Bias | -0.4 | 0.2 | 0.1 | 0.5 | -0.3 | 0.8 | -0.9 | 2.2 | 0.2 | 0.1 |
| RMSD | 0.7 | 0.7 | 0.6 | 0.6 | 0.7 | 1.9 | 1.7 | 3.0 | 1.2 | 1.6 |
| STD | 0.4 | 0.5 | 0.4 | 0.4 | 0.4 | 1.1 | 1.3 | 2.4 | 0.9 | 1.1 |
| r | 0.99 | 0.99 | 0.99 | 0.99 | 0.99 | 0.93 | 0.94 | 0.81 | 0.97 | 0.95 |

and IASI as well as for the reanalyses ERAI and ERA5. Vice versa CFSR and with lesser degree MERRA2 underestimate IWV over sea ice and overestimate over the ocean as well as over Scandinavia and Siberia.

MIRS shows no major anomaly over ocean areas and just a slight negative difference over sea ice in June. However, in May the sea ice edge close to Svalbard becomes rather prominent separating the negative (sea ice) and positive (ocean) bias.







**Figure 9.** Relative difference between the reanalyses median (bottom right; CFSR, ERA-Interim, JRA-55, MERRA2) and the individual products over the study area for June, 2017. The monthly mean sea ice edge for 15 % sea ice concentration derived by AMSR is shown as black dash-dotted line. Central Arctic (dark blue) and Open Ocean (green) regions are marked.

Particularly striking in MIRS is a strong moist anomaly in the south-eastern corner of the study area where no other products show any spatial anomalies. The positive difference is even stronger in May (> 30 %) extending well into Scandinavia. The reason might be that snow melting changes the emissivity in these regions in that season. For completeness, we also consider the





MODIS monthly product as this has been used for IWV studies in the Arctic (Alraddawi et al., 2017). MODIS underestimates IWV nearly everywhere except certain oceanic regions. As MODIS can only retrieve over bright surfaces (only sun glint over ocean), the sampling is rather poor and small scale variations occur which are not evident in the other products.


The smooth spatial structures in all products (except MODIS) clearly indicate that weather related patterns are averaged out on the monthly scale. Biases associated with certain surface types / regions are the dominating uncertainty factors within the different products. To better assess this issue, we compare the mean values of all products for the two selected areas, i.e. the central Arctic and open ocean (cf. Fig. 1) as well as for the closest grid point to Ny-Ålesund (Table 3). During May, when sea ice still persists, reanalyses agree rather well for the central Arctic with mean values between 4.4 (CFSR) and 4.6 $\mathrm{kgm}^{-2}$ (ERAI, ERA5). With the exception of AMSR, satellite products give rather close results with mean values between 4.3 (MIRS) and 4.9 $\mathrm{kgm}^{-2}$ (GOME-2). The situation changes for June when reanalyses have a maximum difference of 1.7 $\mathrm{kgm}^{-2}$ and satellite products of 2.9 $\mathrm{kgm}^{-2}$. The reason for this strong difference of about 20% compared to only 4 % in May is likely to arise from the melting and transformation of the sea ice affecting air-sea fluxes and the difficulty to capture the moist intrusions into the Arctic in space and time. For satellite retrievals, the transformation of the sea ice has even stronger consequences as surface reflectivity and emissivity drastically change leading to differences of about 30%.



For the Open Ocean domain, differences in the IWV monthly means of the reanalyses are around 10 % (slightly higher in May) (Table 3). As already identified by Fig. 8, AMSR, IASI and MIRS are rather similar as microwave retrievals work best over open ocean. AIRS is not too far off but the strong underestimation by GOME-2 needs to be better understood and reduced.

Ny-Ålesund with its suite of ground-based instrumentation is well suited to look deeper into the differences in mean IWV. When looking at the ground-based reference data, also differences in their monthly estimates appear. As discussed before radiosondes have more limitations due to their poorer temporal sampling. Based on the good agreement of MWR to radiosonde data we consider it as reference here. GNSS underestimates mean IWV by roughly 10 % but this is still in the range of the reanalyses and satellite products. ERAI and MERRA2 are both very close to the MWR (differences <0.1 $\mathrm{kgm}^{-2}$) in both May and June. From the satellite products, AIRS and MIRS agree perfectly with the MWR in May, while IASI is closest in June. This illustrates the difficulty to draw a solid conclusion on product quality and certainly longer data records need to be considered.



To put the identified differences into perspective, we analyse how they translate to differences in longwave downward radiation (LWD). Following the approach by Ghatak and Miller (2013), a functional relationship between monthly mean IWV measured by the MWR and LWD as measured by the Baseline Surface Radiation Network station (Maturilli et al., 2015) was derived for Ny-Ålesund (not shown). While there is some scatter mainly arising from different cloudiness in individual months, a clear relation with a nearly linear shape for low IWV saturating for higher IWV values around 15 $\mathrm{kgm}^{-2}$ is present. According to this relation, the difference in IWV satellite products for Ny-Ålesund of about 1.9 $\mathrm{kgm}^{-2}$ evident in May relates to a difference of about 20 $\mathrm{Wm}^{-2}$ in LWD while the stronger IWV difference in June of 2.9 $\mathrm{kgm}^{-2}$ implying a LWD difference of about 25 $\mathrm{W\,m}^{-2}$. This demonstrates the importance to improve the accuracy of estimates at the lower IWV end.







**Table 3.** Monthly mean IWV (in $kgm^{-2}$) for May and June 2017 derived for the areal averages of the Central Arctic and Open Ocean. The mean has only been calculated from those grid points which have valid entries for all products. N denotes the percentage of valid grid points in terms of their number and taking the area dreaction into account. For Ny-Ålesund (NYA) the closest grid point is shown.

| | May 2017 | | | June 2017 | | |
|---|---|---|---|---|---|---|
| | Central Arctic | Open Ocean | NYA | Central Arctic | Open Ocean | NYA |
| N in % | 31.1 / 42.4 | 75.6 / 75.5 | | 39.8 / 53.0 | 94.4 / 94.3 | |
| CFSR | 4.4 | 8.3 | 5.9 | 8.2 | 12.5 | 11.3 |
| ERAI | 4.6 | 7.4 | 5.6 | 7.4 | 11.4 | 10.7 |
| JRA55 | 4.5 | 7.2 | 4.9 | 9.1 | 11.1 | 10.0 |
| MERRA2 | 4.5 | 7.8 | 5.6 | 8.9 | 12.1 | 10.7 |
| ERA5 | 4.6 | 7.4 | 5.4 | 7.4 | 11.3 | 10.7 |
| AIRS | 4.5 | 6.9 | 5.7 | 9.5 | 10.4 | 9.4 |
| AMSR-2 | 6.3 | 7.9 | 6.6 | 11.4 | 12.3 | 12.7 |
| GOME-2 | 4.9 | 4.8 | 4.7 | 10.0 | 9.3 | 8.8 |
| IASI | 4.5 | 7.7 | 5.9 | 9.0 | 11.7 | 10.9 |
| MIRS | 4.3 | 8.1 | 5.6 | 8.5 | 11.6 | 9.4 |
| MODIS | – | – | 6.8 | – | – | 11.1 |
| GNSS | | | 4.9 | | | 9.9 |
| MWR | | | 5.6 | | | 10.8 |
| RS | | | 5.9 | | | 11.2 |

## 5 Conclusions and Outlook

The role of water vapor in Arctic amplification is still poorly understood partly due to the lack of a solid observational data base. Radiosondes launched at only few stations are still considered as the best climatological record to assess trends although depending on the used radiosonde type at each station issues about their uncertainty exist. From their limited data record as well as from global reanalyses only few regions and seasons with robust IWV trends can be derived (Rinke et al., 2019). With the emergence of new satellite series providing already decade long data as well as high resolved reanalysis (ERA5) there is hope for a better assessment of Arctic IWV changes. The ACLOUD/PASCAL period in May/June 2017 performed in the Arctic North Atlantic is exploited to investigate the performance of satellite products as well as reanalyses from instant measurements up to monthly means.

Polar orbiting satellites provide good sampling at high latitudes such that measurements are typically available every hour with maximum gaps of five hours (Fig. 3). Nevertheless, when trying to evaluate satellite products with radiosondes only available at synoptic times sampling is relatively coarse such that a certain satellite only matches with a limited number of





stations. Comparing the performance of different satellite instruments is thus difficult as each instrument "sees" a different set of stations. This gets even more complicated as the quality of radiosondes is likely not comparative across the regions

(Ingleby, 2017). For example, there is an indication that stations in the eastern part of the region underestimate IWV while stations further to the west overestimate. Therefore, in addition to the standard radiosondes we make use of the high quality ACLOUD/PASCAL radiosondes launched from the Polarstern frozen into the ice and enhanced launch activity at Ny-Ålesund. For the latter, comparisons over the ocean reveal the best performance by AMSR (RMSD=0.6 $\mathrm{kgm}^{-2}$) which is not surprising as low frequency microwave measurements are rather directly related to IWV over ocean even in cloudy conditions. However,

over ice AMSR shows a pronounced bias. Here, IASI shows highest skill (RMSD=0.9 $\mathrm{kgm}^{-2}$) which also is true when all radiosonde stations are considered together (RMSD=1.3 $\mathrm{kgm}^{-2}$). The fact that the IASI performance is much better than the one by the similar AIRS instrument is attributed on one hand to the utilisation of collocated high-frequency microwave observations from MHS and MHS, which enables useful sounding in most cloudy conditions, and on the other hand to differences in the retrieval strategy. MIRS which is mainly depending on MHS measurements has good coverage as four different satellites

are used but an underestimation of high IWV values is evident at Polarstern, Ny-Ålesund and radiosonde stations in the north western part of the region while an overestimation for certain Russian stations occurs. Spatial analysis reveals that the overestimation by MIRS (compared to the reanalyses median) extends over a wider region and even far into Scandinavia in May. Therefore this feature could also be related to an insufficient description of the surface which is undergoing melting of snow at that time of the year.

Water vapor in the Arctic is especially difficult to assess as it is prone to high space and time variability. In case of AR events, changes of more than 100 % can occur within a few hours (Fig. 4). As the reanalyses are mainly anchored by radiosonde in the Arctic, the forecast model becomes important in representing the spatio-temporal development of IWV. This leads to differences of around 10 % in daily mean values (Fig. 6) compared to the reanalyses mean, an effect which is even stronger for the different satellite products due to the less frequent orbits. Nevertheless, we can show that the density of satellite overpasses

is high enough for all products - with the exception of MODIS - to smooth out weather related features within monthly mean IWV.

Overall, in agreement with the radiosonde comparison the performance by IASI (as judged in comparison to the reanalyses median) is best on daily as well as on the monthly scale. AIRS, whose products here are generated from infrared-only measurements, displays less accurate IWV than the IASI products. Over open ocean, the low frequency microwave product by AMSR

has even slightly less scatter than IASI but shows a slight bias. The bias is much stronger over sea ice in the central Arctic. However, efforts to improve the retrieval in respect to surface emissivity are ongoing. GOME-2 performs well in May over sea ice but strong overestimation (over Greenland) and underestimation (over ocean) is apparent.

The reanalyses median was composed by the classical global reanalyses CFSR, ERAI, JRA55 and MERRA2. It is interesting to see that the high resolution reanalysis ERA5 is not more similar to ERAI than any other reanalyses on the daily scale. In

this way, a more detailed analysis could identify the role of data assimilation for the positive (ERAI) and negative (ERA5) anomalies close to Nova Zemlja. Similarly, data assimilation of IASI (and MIRS) might be the reason for the similar negative bias of ERA5 in the North Atlantic which is not evident in ERAI. On the monthly scale, ERAI and ERA5 are rather similar



with positive anomalies over the central Arctic and negative anomalies for the rest of the region hinting at similar treatment of surface fluxes. In the central Arctic, strong differences of 30 % in IWV monthly means between satellite products occur in the month of June which likely result from the difficulties to consider the complex and changing surface characteristics of the melting ice within the retrieval algorithms. There is hope that the detailed surface characterization performed as part of the recently finished Multidisciplinary drifting Observatory for the Study of Arctic Climate (MOSAiC) expedition will foster the improvement of future retrieval algorithms.

*Data availability.* The ERA5 hourly data are obtained from https://cds.climate.copernicus.eu/cdsapp#!/dataset/reanalysis-era5-single-levels?tab=form(1). The CFSR, ERAI, JRA-55 and the MERRA2 reanalysis data sets are the same ones as used in (Rinke et al., 2019). AIRS data are provided by NASA's Goddard Earth Sciences Data and Information Services Center (GESDISC; https://disc.gsfc.nasa.gov/; AMSR and GOME-2 data are available from IUP Bremen by request. IASI data were downloaded as orbital data from http://archive.eumetsat.int/usc/; the orbital IWV product (NetCDF4 Swath files level2a (SND, IMG)) is obtained from the NOAA online database (https://www.bou.class.noaa.gov). MODIS data are obtained from the National Aeronautics and Space Administration (NASA) online database (https://ladsweb.modaps.eosdis.nasa.gov/, monthly mean data are retrieved from https://neo.sci.gsfc.nasa.gov/view.php?datasetId=MYDAL2_M_SKY_WV&date=2017-12-01. HATPRO data from Polarstern are available at (Griesche et al., 2019).

*Author contributions.* SC prepared the manuscript with contributions from all authors. KE, TN and MMa prepared the Ny-Ålesund analysis. MMe supported the overall satellite comparison. AR investigated the reanalyses (CFSR, ERAI, JRA55, MERRA2). DS prepared the direct intercomparison with radiosondes. PK performed the spatial product analysis. CV and IG investigated the atmospheric river events. SN developed the AMC-DOAS retrieval method and analysed the GOME-2 data product. RS, ATG, GS prepared the AMSR analysis. TA contributed the IASI analysis. MS provided the context with GVAP.

*Competing interests.* The authors declare that they have no conflict of interest.

*Acknowledgements.* We gratefully acknowledge the funding by the Deutsche Forschungsgemeinschaft (DFG, German Research Foundation) – Project-ID 268020496 – TRR 172, within the Transregional Collaborative Research Center "ArctiC Amplification: Climate Relevant Atmospheric and SurfaCe Processes, and Feedback Mechanisms (AC)3". MS acknowledges the financial support by the EUMETSAT member states through CM SAF. We thank GFZ Potsdam for providing the GNSS measurements and Hannes Griesche from Tropos for the HATPRO measurements from Polarstern.

# 6 Appendix



**Figure A1.** Joint distribution of daily means from the reanalyses (x-axis: CFSR, ERA5, ERA-Interim, JRA-55, MERRA2) and reanalyses median (y-axis CFSR, ERA-Interim, JRA-55, MERRA2) for Central Arctic (left), Open Ocean (middle) and the full region (right). The time period is May to June 2017. The color indicates the relative fraction of the IWV.





**Figure A2.** Relative difference between the reanalyses median (bottom right; CFSR, ERA-Interim, JRA-55, MERRA2) and the individual products over the study area for May, 2017. The monthly mean sea ice edge for 15% sea ice concentration derived by AMSR is shown as black dash-dotted line.





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



**Table A1.** Skill scores for the intercomparison of individual radiosonde stations to satellite IWV in May and June 2017 for all valid data pairs in terms of bias, STD and RMSD (all in $kg\,m^{-2}$) and correlation coefficient (r). N denotes the number of samples.

| Instrument | | danmark | ittoqqor | ammassalik | summit | jmayen | polarstern | nyalesund | andoya | bjornoya | lulea | sodankyla | kandalaksa | murmansk | sojna | karmakuly | narian | pechora | krenkelja | allstations |
|---|---|---|---|---|---|---|---|---|---|---|---|---|---|---|---|---|---|---|---|---|
| AIRS | N | 34 | 21 | - | 29 | 18 | 17 | 39 | 31 | 25 | 10 | 31 | 33 | 29 | 28 | 26 | 30 | 28 | 32 | 461 |
| | r | 0.89 | 0.86 | - | 0.92 | 0.92 | 0.94 | 0.90 | 0.95 | 0.89 | 0.92 | 0.92 | 0.92 | 0.94 | 0.88 | 0.96 | 0.90 | 0.93 | 0.83 | 0.92 |
| | BIAS | -0.1 | 1.0 | - | -0.1 | -1.8 | -0.6 | -0.4 | 0.4 | -0.8 | 1.0 | -0.4 | -0.6 | -0.6 | -0.4 | 0.1 | 0.8 | -2.3 | 0.7 | -0.3 |
| | STD | 1.3 | 1.7 | - | 0.4 | 1.9 | 1.4 | 1.7 | 1.5 | 1.5 | 2.2 | 1.9 | 1.8 | 1.4 | 2.1 | 1.4 | 1.9 | 2.5 | 1.5 | 1.9 |
| | RMSD | 1.3 | 1.9 | - | 0.4 | 2.6 | 1.5 | 1.7 | 1.5 | 1.6 | 2.3 | 1.9 | 1.9 | 1.5 | 2.1 | 1.4 | 2.1 | 3.4 | 1.7 | 1.9 |
| AMSR | N | 37 | 35 | - | - | 18 | 17 | 49 | 35 | 33 | 16 | - | - | - | 34 | 31 | - | - | 24 | 329 |
| | r | 0.89 | 0.89 | - | - | 0.99 | 0.96 | 0.99 | 0.99 | 0.97 | 0.89 | - | - | - | 0.83 | 0.98 | - | - | 0.75 | 0.91 |
| | BIAS | 2.7 | 2.8 | - | - | 1.0 | 3.3 | 0.9 | 1.1 | 1.0 | 1.7 | - | - | - | 0.8 | 1.3 | - | - | 3.1 | 1.7 |
| | STD | 1.6 | 1.6 | - | - | 0.7 | 0.9 | 0.7 | 0.7 | 0.9 | 2.1 | - | - | - | 3.0 | 1.0 | - | - | 1.8 | 1.7 |
| | RMSD | 3.1 | 3.2 | - | - | 1.2 | 3.4 | 1.1 | 1.3 | 1.4 | 2.6 | - | - | - | 3.1 | 1.6 | - | - | 3.5 | 2.4 |
| GOME | N | 48 | 54 | 27 | 34 | 30 | 24 | 79 | 22 | 52 | - | - | - | - | - | - | - | - | 58 | 428 |
| | r | 0.95 | 0.84 | 0.07 | 0.94 | 0.85 | 0.90 | 0.89 | 0.94 | 0.85 | - | - | - | - | - | - | - | - | 0.93 | 0.85 |
| | BIAS | 0.4 | 0.6 | -0.2 | 1.6 | -2.0 | 0.8 | -1.6 | -1.8 | -1.7 | - | - | - | - | - | - | - | - | 1.0 | -0.3 |
| | STD | 1.1 | 1.5 | 3.7 | 0.9 | 2.1 | 1.6 | 1.8 | 1.6 | 2.0 | - | - | - | - | - | - | - | - | 1.0 | 2.2 |
| | RMSD | 1.2 | 1.7 | 3.7 | 1.8 | 2.9 | 1.8 | 2.5 | 2.4 | 2.6 | - | - | - | - | - | - | - | - | 1.4 | 2.2 |
| IASI | N | 60 | 57 | 111 | 112 | 31 | 27 | 81 | 60 | 61 | - | - | - | - | - | 48 | - | - | 60 | 708 |
| | r | 0.97 | 0.88 | 0.91 | 0.90 | 0.97 | 0.98 | 0.97 | 0.95 | 0.98 | - | - | - | - | - | 0.96 | - | - | 0.92 | 0.96 |
| | BIAS | -0.6 | -0.1 | -1.5 | 0.1 | -0.6 | -0.2 | -0.7 | -0.5 | 0.1 | - | - | - | - | - | 0.7 | - | - | 0.8 | -0.3 |
| | STD | 1.0 | 1.3 | 1.5 | 0.4 | 1.2 | 0.9 | 1.1 | 1.4 | 0.8 | - | - | - | - | - | 1.0 | - | - | 1.0 | 1.3 |
| | RMSD | 1.2 | 1.3 | 2.1 | 0.4 | 1.3 | 0.9 | 1.3 | 1.5 | 0.8 | - | - | - | - | - | 1.2 | - | - | 1.3 | 1.3 |
| MIRS | N | 57 | 54 | 53 | 51 | 30 | 32 | 95 | 38 | 59 | - | 37 | 35 | 36 | 9 | 58 | 26 | 44 | 82 | 787 |
| | r | 0.88 | 0.71 | 0.75 | 0.96 | 0.94 | 0.90 | 0.90 | 0.97 | 0.97 | - | 0.93 | 0.88 | 0.94 | 0.87 | 0.89 | 0.84 | 0.90 | 0.87 | 0.88 |
| | BIAS | -0.3 | 0.0 | -0.1 | 0.1 | -0.7 | -1.4 | -2.2 | -1.0 | 0.1 | - | -0.1 | 1.4 | 1.3 | -0.1 | 0.5 | 2.3 | 1.6 | 0.1 | -0.1 |
| | STD | 1.5 | 1.9 | 2.0 | 0.3 | 1.7 | 1.8 | 2.3 | 1.7 | 1.0 | - | 2.2 | 2.1 | 1.3 | 2.6 | 1.8 | 1.6 | 2.4 | 1.3 | 2.1 |
| | RMSD | 1.5 | 1.9 | 2.0 | 0.3 | 1.8 | 2.3 | 3.2 | 2.0 | 1.0 | - | 2.2 | 2.5 | 1.8 | 2.5 | 1.9 | 2.7 | 2.8 | 1.3 | 2.1 |