# Peer review of "A systematic assessment of water vapor products in the Arctic: from instantaneous measurements to monthly means"

_Atmospheric Measurement Techniques, 2020_

## Author Comment (AC1)

**Reply to Reviewer 1**

We thank the reviewer for his/her constructive comments which helped to improve the quality of the paper. In the following reviewer comments are marked in italics.

*"I enjoyed reading this paper and am interested in the results. I think a little more care should be given to how the vertical layering is defined for each product since past experience has shown that methodology differences among the products is a significant contributor to IWV variations among products. With the possible exception of AMSR2, none of these IWV estimates are direct measurements, they are all derived through integrating the atmosphere vertical column. The problem with that is of course the number density of water vapor molecules decreases exponentially with height. That means that the largest fraction of IWV is within the lowest 100's of meters of the atmosphere. Even small differences in the vertical layering used can change the estimated IWV. This issue may have been addressed in the project but since it is hardly mentioned in the paper I have my doubts that it was done carefully enough. However, despite that caveat I feel the results warrant publication as is."*

The reviewer is right that the layering of the atmosphere plays a crucial role. In fact, even different IWV estimates (with differences of 1mm or even more) are available for the same radio sounding depending whether high vertical resolution, pressure and significant levels (such as in IGRA) or pressure levels are used in the integration. Strongest differences occur often when rapid moisture changes with height occur, e.g. in moisture inversions which are frequent in the Arctic. We added a paragraph to discuss this issue.

"For most of the products IWV is not a directly measured quantity but derived by integrating the vertical humidity profile. In this exercise differences between products can occur due to differences in the vertical sampling and the definition of the lower/upper boundary. The first point is of special relevance for radiosonde measurements and model profiles when strong vertical moisture gradients occur, e.g. during moisture inversions which are frequent in the Arctic (Naaka et al., 2018). The first effect can lead to differences between high-resolution radio soundings and those only using main pressure levels of several $kgm^{-2}$ for individual profiles. The second effect mainly concerns the lower boundary as a height difference between two products can cause systematic biases and is most important in orographically structured terrain where the effective footprint of models and satellite products causes different average elevations. As a rule of thumb a height difference of 100 m in the presence of 5 $gm^{-3}$ absolute humidity (typical maximum for the Arctic) causes an IWV difference of 0.5 $kgm^{-2}$. Similarly, synoptic pressure deviations can be problematic (Divakarla et al., 2006) when vertical profiles are provided on fixed pressure grids."

**Specific Comments:**

*1. This paper uses the word IASI for three different meanings; the sensor, the radiances, and the EUMETSAT L2 product. Strongly recommend the following:*

*"IASI" for the sensor*

*"IASI radiances" for the infrared spectrum subset assimilated into reanalysis products*

*"IASI L2 PPFv6" for the EUMETSAT L2 v6 product*

We modified the text accordingly. However, in order to avoid too complex figures we kept the simplified annotation in the figures.

*2. In a similar way the AIRS product should be referred to as "AIRS L2 v6 IR-Only" to distinguish it from many AIRS products commonly in the past and in the future.*

*It is not really fair to compare what you call "AIRS" to "IASI" when the Aqua satellite was in it's 15th year of operation in 2017 after both the AMSU and MHS have failed. A fair comparison of AQUA to METOP products can be found in Roman et al. (2016) when all sen correctly. While there were differences between the NASA and EUMETSAT products, they are not as serious as those in this paper which is using the AIRS IR-only product. I just want you to appreciate that it's not a fair comparison and that if you don't want to remove the AIRS results you should change what you say about them. For example I think you can use AIRS IR-Only as an example of how METOP can expect to degrade with time, or at least avoid implying negative connotations about "AIRS" by replacing "AIRS" with "degraded AQUA sensors" in the conclusions. As you should know the AIRS sensor itself is actually working normally after 19 years.*

The reviewer is completely right that it is not fair to compare our AIRS and IASI products directly. We had hoped that this was already clear from our discussion conclusions. We have now revised the text at several instances and use the term *AIRS L2 v6 IR-Only* to make this more clear including a reference to Roman et al. (2016).

*3. Line 201. "as well By" Punctuation.*
changed

*4. Line 213 (Figure 2). Can the reanalyses difference be explained by a difference in the reanalysis surface pressure compared to the station surface pressure? Any surface pressure difference should be documented.*

For the AR event on 6 June 2017 shown in Fig. 2 all reanalysis agree nearly perfectly with observation for Ny Alesund while for Polarstern differences are more for all reanalysis pronounced. This is likely due to differences in the underlying model as much less observations are assimilated in that area. Another possible reason is the coarse temporal resolution of the reanalyses output (as can be seen in more details on Fig. 4 – the reanalyses missing a peak in IWV because it occurs between the 6-hourly output.

*5. Line 220. Reanalysis.    done*
*Can you include a sentence that describes how the IWV was computed using NWP profiles? The details of that are important to document.*

Integrated water vapor has been directly downloaded from the reanalysis production sites. Therefore, no integration was performed by ourselves. However, this is an important point as we noted ourselves in another study that the integration just over water vapor at pressure levels (as frequently done) gives slightly different values than integrating over all model levels. This is now discussed in the text – see answer to your general comment.

*6. Line 253. I don't see the GNSS symbols in the lower panel of Figure 2. Can you include some explanation? Perhaps there is no GNSS at the Polarstern location? If true please clarify in the caption.*

Indeed there is no GNSS at Polarstern. We changed the figure caption accordingly.

*7. Line 286. add ", with a station elevation of 30m."*
changed

*8. Line 286. Should estimate the PWV of the 30m Ny-Alesund station elevation and include that number in the text of the paper since it would contribute to a small bias relative to the ocean satellite observations.*

We wonder there the reviewer found the value of 30 m as the station height is 11 m above mean sea level (asl)? Even with 30 m and maximum absolute humidity amounts to about 10 $gm^{-3}$ this would lead to an underestimation of the water vapor column by only 0.03 $kgm^{-2}$ compared to msl which is negligible. More importantly, the satellite measurements over land are also affected by stronger orography as the satellite footprint covers a wider area. In response to your general comment we added a paragraph on the vertical height differences (see above) and also mention station height and height of the closest mountain.

9. Line 345.  Did you compare the AIRS and IASI surface pressure to the station pressure?  Since AIRS and IASI retrievals use NWP surface pressure as input the derived IWV can often contain an error due to a bias in assumed surface pressure in the satellite retrieval. This issue has been known for a long time so I am surprised it is not mentioned as a contributing factor in this paper.
"For both the land and sea cases, uncertainties in the specification of surface pressure for the retrievals through interpolation of NCEP_GFS surface pressure (the only ancillary parameter in the AIRS APS) might be causing an error term, which requires further investigation. "
https://agupubs.onlinelibrary.wiley.com/doi/full/10.1029/2005JD006116

**Inspired by the general comment of the reviewer we added a general paragraph on the influence of vertical layering. Note that IASI has a fixed pressure grid (with 101 levels) to describe the humidty profile and that the distance between levels at the surface is about 25 hPa which is more than any difference in surface pressure.**

*10. Line 503.  This is nice. I like how you convert to surface IR flux. I agree this is the relevant issue.*

**See the simple sketch below. Because a similar type of drawing can be found in the reference (Ghatak and Miller, 2013) we refrain from including it into the paper.**

[Figure]

*11. Line 533. MHS is repeated*
*"from MHS and MHS"*

corrected

*12. Line 556. This statement about ERA5 and IASI is incorrect and should be modified or removed. ERA5 does not assimilate L2 sounding products it only assimilates a radiances. Also it is nearly impossible to attribute ERA5 changes to any specific data inputs. Since this is assertion is not proven in the paper it should not be in the conclusions.*

We modified the sentence to "It is interesting to see that the high resolution reanalysis ERA5 is not more similar to ERAI than any other reanalyses on the daily scale. However, identifying the reasons behind the differences is not straightforward as changes in the underlying model and in data assimilation can play a role."

---

## Author Comment (AC2)

**Reply to Reviewer 2**

We thank the reviewer for his/her constructive comments which helped to improve the quality of the paper. In the following reviewer comments are marked in italics.

*The paper addresses the very interesting and hot topic of water vapour concentration and distribution in the arctic regions with different satellite instruments and retrieval algorithms. Both L2 and L3 products are considered, which are compared with reference data derived from various re-analyses, radiosonde and GNSS observations. I have found the paper well written and complete; therefore, my remarks are minor.  Here a few comments to improve the reading and understanding of results.*

1.  *For IASI and MIRS, the L2-product accuracy is not given while it has to be provided. It could explain in part the diverse performance of the two products. Based on the paper results, it seems that the bias of MIRS has not been well assessed previously.*

The quality of the IASI L2 sounding products (and also for MIRS) varies with the scenes. Better performances are expected in clear sky and with larger thermal constrasts. Assessments against correlative radiosondings and intercomparison to numerical models stratified in quality ranges as per IASI L2 quality indicators confirmed that biases (systematic errors) are close to zero. The precision (random component of the uncertainties) is better than 1K for temperature in the free troposphere in the best retrieval classes. The sounding precision decreases nearer to the surface. The exact characterisation of IASI L2 precision is more difficult as the collocation uncertainties with radiosondes (who most of the time are distant by at least 1 to 2h in time) are expected to play a larger role in the IASI-sonde departure budget. Precision of about 1.5K are typically expected. Same considerations apply to humidity, but the uncertainties in absolute and relative terms vary with the actual moisture load. Thus, it is not possible to give an easy answer. Note, that we cite the results of the intensive intercomparison of IASI and AIRS to GNSS measurements by Roman et al., 2016.

2.  *The AR event should be better defined for the benefit of the reader*

We also modified some text for clarity and added the following definition: " ARs have been also associated with several cyclones helping the moisture supply within the same AR structure (Sodemann and Stohl, 2013). In the Polar regions ARs are often associated with moisture inversions showing maxima in specific humidity between 800 and 900 hPa (Gorodetskaya et al., 2020). Here we choose the AR event from 6 June 2017 at 12 UTC to illustrate the capabilities  of  the  different products (Fig. 4). Note, that  this  is  only  one  out  of  three  AR  events  that  occurred  during ACLOUD/PASCAL documented in detail by Viceto et al, (submitted)."

3.  *The IASI retrievals come from a combination of IR and MW instruments. The product simply as IASI is confusing; maybe the authors could use the acronym IASI/AMSU/MHS, which could even be shortened to IAM.*

This concern was also addressed by reviewer 2. We followed his/her naming suggestion.

4.  *Given the high cloud coverage of the arctic, IASI (the IR ) is expected to yield a very poor contribution in comparison to AMSU and MHS; therefore, the large performance difference between IASI and MIRS (based again on AMSU and MHS) is not understood and would deserve a bit of more explanation.*

As shown by Knudsen et al. (2018) average cloud top height during ACLOUD is about 700 hPa showing the dominance of low level clouds. Specific to the Arctic are also humidity inversions leading to high moisture amounts in the mid-troposphere which can feed clouds from above. In fact, during the AR on 6 June specific humidity is maximum at around 800 hPa. Therefore, IASI

still provides important information on mid-level moisture which due to its more pronounced weighting functions at these altitudes provides additional information compared to microwave only retrievals. Thus, the synergistic exploitation of microwave observations from AMSU and MHS together with the hyperspectral sounding from IASI enable quasi nominal sounding down to the cloud top and to preserve relatively good performances in most cloud-effected pixels. The assessments carried out with reference radiosonding confirmed that the quality indicator (= uncertainty estimates in the lower tropo) is a reliable. User can rely on this indicator to perform quality control tailored to their needs.  A short discussion was added in the conclusion.

5.   *On the same subject, what were the sky conditions during the ice camp in June? Clear Sky? Clear sky, in fact, would in part explain the better performance of IASI in comparison to MIRS.*

The monthly mean cloud fraction at Ny-Alesund for May 2017 is 75% and in June 83% (Nomokonova et al., 2019, figure 9).  For the ice camp of Polarstern it is 88 %. The information has been added to the paper.

6.   *By the way, with reference to Fig.6 (6 June 2017), the authors say that the same bias pattern seen in IASI and MIRS is because they both make use of MW units. But this is conflicting with the MIRS large bias seen in Fig. 2. Once again, the MIRS bias could be better explained in light of the IASI good performance.*

In Fig. 6 the difference seen between the different products ("biases") occurs mostly due to synoptic systems which are shifted in the different products due to their spatio-temporal sampling. This "weather noise" is much higher than the systematic long-term effects – compare Figure 6 and 9. Because the METOP satellites fly IASI and two of the MIRS instruments their sampling is rather similar leading to the same spatial difference patterns. The MIRS bias seen in Fig.2 especially in June is on the order of a few kgm-2 while the daily means in Fig.6 have differences between +50 and – 20 % (for IASI) which is up to 10 kgm-2. In fact, if you look at the area of highest relative differences close to Svalbard in Fig.6 you can see that IASI reaches up to 50 % while for MIRS it is up to 20-30 %. Therefore we do not see any contradiction.

7.   *Tab A1 Skill scores. Why is the GOMe r for Ammasalik 0.07? This is really an outlier with respect to the r for other stations. Is that correct*

The low correlation for this station is mainly due to a few measurement points where the GOME-2 WV is considerably lower than the RS WV. Possible reasons for this are the influence of clouds and changing surface albedo due to e.g. changing ice cover.

---

## Author Response (AR2)

**Reply to 2nd Review**

I have only one additional minor revision; Line 237: Change "reanalysis" back to "reanalyses". I'm not sure why this was changed during revisions."

We corrected this.